# Neurofilament Light Chain Levels in Serum and Cerebrospinal Fluid Do Not Correlate with Survival Times in Patients with Prion Disease

**DOI:** 10.3390/biom15010008

**Published:** 2024-12-25

**Authors:** Mika Shimamura, Kong Weijie, Toshiaki Nonaka, Koki Kosami, Ryusuke Ae, Koji Fujita, Taiki Matsubayashi, Tadashi Tsukamoto, Nobuo Sanjo, Katsuya Satoh

**Affiliations:** 1Biomedical Research Support Center, Nagasaki University, 1-12-4 Sakamoto, Nagasaki 852-8523, Japan; shima-m@nagasaki-u.ac.jp; 2Unit of Medical and Dental Sciences, Department of Health Sciences, Nagasaki University Graduate School of Biomedical Sciences, 1-12-4 Sakamoto, Nagasaki 852-8523, Japan; wj2023nagasaki@gmail.com; 3Division of Cellular and Molecular Biology, Nagasaki University Graduate School of Biomedical Sciences, 1-12-4 Sakamoto, Nagasaki 852-8523, Japan; 4Division of Public Health, Center for Community Medicine, Jichi Medical University, Tochigi 329-0498, Japan; k.kosami@jichi.ac.jp (K.K.); shirouae@jichi.ac.jp (R.A.); 5Department of Neurology, Tokushima University Graduate School of Biomedical Sciences, 3-18-15 Kuramoto-cho, Tokushima 770-8503, Japan; kfujita@tokushima-u.ac.jp; 6Department of Neurology and Neurological Science, Tokyo Medical and Dental University, Graduate School of Medical and Dental Sciences, 1-5-45 Yushima Bunkyo-ku, Tokyo 113-8510, Japan; taiki.matsubayashi55135@gmail.com; 7Department of Neurology, National Center of Neurology and Psychiatry (NCNP), 4-1-1 Ogawa-Higashi, Kodaira, Tokyo 187-8551, Japan; tukamoto@ncnp.go.jp; 8Department of Internal Medicine, Division of Neurology, Kudanzaka Hospital, 1-6-12 Kudan-minami, Chiyoda-ku, Tokyo 102-0074, Japan; n-sanjo.nuro@tmd.ac.jp; 9Department of Neurology and Neurological Science, Tokyo Medical and Dental University, Graduate School of Medical and Dental Sciences, 2-12-1 Ookayama, Meguro-ku, Tokyo 152-8550, Japan; 10Unit of Brain Research Core Unit, Leading Medical Research Unit, Nagasaki University Graduate School of Biomedical Sciences, Nagasaki 852-8523, Japan

**Keywords:** neurofilament light chain, prion, Creutzfeldt–Jakob disease, serum, cerebrospinal fluid

## Abstract

Prion diseases, including Creutzfeldt–Jakob disease (CJD), are deadly neurodegenerative disorders characterized by the buildup of abnormal prion proteins in the brain. This accumulation disrupts neuronal functions, leading to the rapid onset of psychiatric symptoms, ataxia, and cognitive decline. The urgency of timely diagnosis for effective treatment necessitates the identification of strongly correlated biomarkers in bodily fluids, which makes our research crucial. In this study, we employed a fully automated multiplex ELISA (Ella^®^) to measure the concentrations of 14-3-3 protein, total tau protein, and neurofilament light chain (NF-L) in cerebrospinal fluid (CSF) and serum samples from patients with prion disease and analyzed their link to disease prognosis. However, in North American and European cases, we did not confirm a correlation between NF-L levels and survival time. This discrepancy is believed to stem from differences in treatment policies and measurement methods between Japan and the United States. Nonetheless, our findings suggest that NF-L concentrations could be an early diagnostic marker for CJD patients with further enhancements. The potential impact of our findings on the early diagnosis of CJD patients is significant. Future research should focus on increasing the number of sCJD cases studied in Japan and gathering additional evidence using next-generation measurement techniques.

## 1. Introduction

Prion diseases are conditions characterized by the accumulation of abnormal prion proteins in the brain, leading to neuronal loss. They arise when the normal prion protein (PrP^C^), which primarily has an α-helical structure, transforms into an abnormal form (PrP^Sc^), characterized by a predominantly β-pleated sheet structure. The most prevalent prion disease is Creutzfeldt–Jakob disease (CJD), with 80–95% of cases being of the sporadic type, 10–15% of the genetic type (mostly familial), and fewer than 1% of the acquired type. CJD typically presents with rapidly progressing dementia [1].

sCJD is found globally, with approximately 1–2 cases per million. Symptoms develop quickly, and the outlook is generally poor. However, survival data reveal differences among countries; patients in Europe and the United States typically survive about 5 months post-onset, whereas patients in Japan tend to live longer. This discrepancy is believed to stem from the longer duration of immobility and mutism in patients [2,3]. The Japanese treatment guidelines state that when a patient experiences akinetic mutism, tube feeding and other symptomatic therapies are implemented to extend life. Nakatani et al. [4] found that the decision to provide tube feeding to a patient with akinetic mutism influences their survival rate, which has notably extended the survival duration of CJD patients in Japan.

In contrast, in Western nations, aggressive symptomatic treatments such as tube feeding and life-extending interventions are often avoided for patients with akinetic mutism, primarily due to economic and ethical considerations [5].

Currently, there is no cure for this disease, which rapidly progresses to dementia and is ultimately fatal. While there are similarities, such as a lengthy incubation period, the clinical symptoms vary widely; therefore, prompt diagnosis soon after the onset is crucial. The standard diagnostic criteria for prion diseases involve a blend of specific symptoms, electroencephalography (EEG), and the detection of 14-3-3 and tau proteins in the cerebrospinal fluid (CSF) [6,7,8,9,10,11]. The RT-QuIC assay, which was developed by Atarashi et al. [12] in 2011, has significantly advanced as an antemortem diagnostic tool for sporadic CJD (sCJD), and its potential for early prion disease diagnosis has garnered increasing attention [13].

Biomarkers linked to diseases are essential for the timely and precise diagnosis of neurodegenerative conditions. CSF envelops the central nervous system. It directly interacts with the extracellular environment of the brain and spinal cord, making it an important indicator of biochemical alterations associated with neurodegenerative diseases. Among the CSF biomarkers, research on Alzheimer’s disease has progressed the furthest, introducing several biomarkers into clinical trials [14]. Despite the benefits of CSF, frequent collection poses challenges due to its invasive and painful nature. Furthermore, the infection risk varies notably, with CSF having higher susceptibility than lower-risk body fluids like blood [15].

As a result, there is increasing interest in creating blood biomarkers for population screening. Neurofilaments are neuron-specific proteins found throughout the central nervous system (including the spinal cord) and the ganglia of the peripheral nervous system. They are key components of the cytoskeleton in axons and dendrites. Neurofilaments consist of four subunits: NF-H, NF-M, neurofilament light chain (NF-L), and α-internexin, with NF-L being the most abundant and soluble, making it a promising biomarker. When nerve cells are damaged by inflammation, NF-L is released from the axons into the bloodstream. This protein, encoded by the NEFL gene, is present in axons and plays a crucial role in maintaining nervous system function. NF-L levels in the CSF and blood increase during axonal injury and nerve cell degeneration, such as in CJD [16]. Previous studies have demonstrated significant correlations between CSF and NF-L levels in Alzheimer’s disease (AD) [17,18,19], and these marker levels are elevated in the prodromal phase of familial AD (fAD). Changes in serum NF-L levels occur in mutation carriers of fAD patients approximately 10 years before symptom onset. Furthermore, serum NF-L can predict disease progression and neurodegeneration in fAD cases and may be a valuable biomarker [17]. Wojdała et al. showed that plasma NF-L, but not serum NF-L, is the most reliable biomarker for AD progression, and their results suggest that measuring plasma NF-L levels may be useful for the early detection of AD, even during the asymptomatic phase. CSF and plasma NF-L may also be important for disease staging and treatment monitoring [18]. Some researchers suggested that the combination of NF-L and phosphorylated tau can predict cognitive decline and AD progression separately [19]. NF abnormalities occur in the lower and upper motor neurons and their axons in patients with amyotrophic lateral sclerosis (ALS) [20,21]. CSF NF-L is highly specific for diagnosing ALS, and serum and plasma NF-L are correlated with mortality [22,23].

Elevated serum and plasma NF-L levels have emerged as significant biomarkers for Parkinson’s disease (PD). In a seminal study, Pilotto et al. demonstrated that plasma NF-L levels were significantly elevated in patients with PD compared with age-matched healthy controls [24]. Furthermore, increased plasma NF-L levels were strongly correlated with the rate of motor function deterioration. These observations provide compelling evidence for the utility of NF-L measurements as a prognostic biomarker, enabling the stratification of patients based on their predicted rate of motor decline at initial assessment [24]. Another study showed that elevated serum NF-L levels are strongly correlated with the progression of PD. Patients with higher serum NF-L levels experienced more rapid deterioration of motor function. Thus, serum NF-L levels are associated with the onset of PD and the subsequent decline in physical capabilities [25].

Research indicates a link between COVID-19 severity, Chronic Fatigue Syndrome, and plasma NF-L concentrations, suggesting that NF-L could serve as an effective biomarker for neurodegenerative diseases and axonal damage [26]. Furthermore, studies have been published in Europe and the United States, indicating that CSF, plasma, and serum NF-L can serve as effective biomarkers [27,28,29]. Given the substantial body of evidence demonstrating correlations between NF-L and various neurological disorders, CSF- and blood-based biomarkers may be valuable tools in early diagnostic protocols in clinical practice.

As noted previously, survival times vary between Japan and the West [2,3], which means that identical outcomes might not be observed. Thus, exploring the connection between the prognosis of sCJD patients in Japan and NF-L chains is crucial.

In this study, we measured NF-L concentrations with a fully automated multiplex enzyme-linked immunosorbent assay (ELISA) Ella^®^ (Bio-Techne, Minneapolis, MN, USA), using CSF and serum from patients diagnosed with prion disease. We thoroughly analyzed the findings of this study alongside the previously collected results regarding 14-3-3 protein levels, total tau protein amounts, and an RT-QuIC assay conducted in our laboratory. This comprehensive review aims to determine whether NF-L concentrations in serum and CSF are linked to patient prognosis.

## 2. Materials and Methods

### 2.1. Human Samples

A total of 72 patients were randomly selected for further investigation. The Japanese Surveillance Committee verified the diagnoses and subsequent analyses of all patients. Patient consent was obtained in accordance with the guidelines of the Declaration of Helsinki. This study was approved by the Institutional Ethics Committee of Nagasaki University Graduate School of Biomedical Sciences (reference number 23042804-2. The following documents were approved on 30 September 2019 and reapproved on 28 April 2023 for prognostic purposes).

### 2.2. Biochemical Analysis of CSF Samples (Total Tau [T-Tau] Protein via ELISA; 14-3-3 Protein Using Western Blot and RT-QuIC Assay)

The results for CSF t-tau, 14-3-3 protein, and RT-QUIC were derived from earlier studies conducted in our lab [13,30].

### 2.3. Measurement of Neurofilament Light (NF-L)

The CSF and serum samples were centrifuged at 9000× *g* for four minutes. Following centrifugation, the CSF was diluted fourfold and the serum was diluted eightfold before dispensing them into cartridges. The NF-L levels in both the CSF and serum were measured using an Ella^®^ automated immunoassay system (Biotechne, Minneapolis, MN, USA) with a human NF-L kit (PlasmaSerum: SPCKB-PS-002448, CSF: SPCKB-HF-002988, Biotechne, Minneapolis, MN, USA), following the manufacturer’s guidelines.

### 2.4. Statistical Analysis

All data are expressed as mean ± SD. Simple regression analysis was performed to calculate the correlation coefficients (R^2^) and Pearson’s correlation coefficients. The study used JMP 17.2 for statistical processing.

## 3. Results

### 3.1. Patients’ Profiles

This study included 72 patients (40 males and 32 females) with a mean age of 70.83 (±10.7) years. The analysis of CSF biomarkers revealed a positivity rate of 84% for total tau protein; levels of tau protein at or above 1300 pg/mL were considered positive. The positivity rate of the 14-3-3 protein was 88.9%, and the RT-QuIC assay (second generation) showed a positivity rate of 80.6%. The time from the onset of CJD symptoms to patient mortality varied considerably, ranging from 1 month to 71 months. The mean survival time was 13.37 (±13.2) months, with a median survival time of 6 months, indicating significant variability in disease progression and outcomes (Table 1). These findings underscore the diversity of clinical trajectories in patients with CJD and highlight the potential utility of CSF biomarkers as diagnostic indicators for disease presence and progression. Genetic analyses conducted by the Surveillance Committee Secretariat revealed no discernible genetic mutations. Furthermore, genotyping of the codon 129 polymorphism associated with solitary prion disease demonstrated a homozygous methionine/methionine (MM) genotype.

### 3.2. Serum and CSF NF-L in Prion Disease

The serum NF-L level was 219.28 ± 184.5 pg/mL and the CSF NF-L level was 9912.67 ± 8609.1 pg/mL (mean ± SD). Notably, these quantitative findings were comparable to the values previously reported by Steinacker et al. [28]. The high correlation between methodologies provides solid evidence for the analytical reliability and validity of Ella^®^ for NF-L quantification.

### 3.3. Relationship Between Disease Duration and NF-L Levels

The relationship between the duration from the onset of CJD to death and NF-L concentration was analyzed (Figure 1). The *X*-axis represents disease duration, while the *Y*-axis displays NF-L concentration: A indicates CSF, and B indicates serum. The CSF NF-L and serum NF-L values were similar to those previously reported by Steinacker and colleagues, and minimal differences were observed between Westerners and Japanese individuals [28]. Furthermore, no correlation was detected between the duration of the disease and NF-L values in either CSF or serum (Figure 1A,B). While there was no significant difference between the NF-L values in CSF and serum, a slight correlation trend was observed (correlation coefficient: 0.576) (Figure 1C).

### 3.4. Relationship Between Period from Onset to Akinetic Mutism and NF-L

In Japan, tube feeding is utilized for patients who cannot eat independently, whereas this practice is not commonly adopted in North American and European cases. We believed that the NF-L concentration from the onset to akinetic mutism could be the most relevant for analyzing the Western data (Figure 2). Comparing the average values over different periods revealed no correlations (Figure 2A,B) or significant differences (Figure 2C).

### 3.5. Relationship Between CJD Progression and NF-L Levels

A significant number of Japanese patients with CJD present with classical CJD (MM I), characterized by rapid progression and a brief duration of the disease. We hypothesized that this might account for the discrepancies observed in results from Europe and the United States, prompting our analysis of disease duration categorized into early and long-term phases (Figure 3A–D). Additionally, we assessed disease duration in two groups: those lasting 20 months or fewer and those exceeding 20 months; however, no statistical correlation emerged from these analyses. Moreover, we found no correlation between levels of NF-L in CSF and serum (Figure 3E).

### 3.6. Relationship Between Age at Onset and CSF/Serum NF-L

The relationship between age at onset and CSF and serum NF-L levels was assessed. No statistically significant differences in CSF NF-L levels were detected across the age-at-onset groups. Serum NF-L levels tended to be higher in patients with disease onset at an age of ≥80 years than in those with disease onset at an age ≤ 60 years. However, this difference was not statistically significant (Figure 4A,B). No statistically significant correlations were detected in either the younger or older age-at-onset groups (Figure 4C,D).

## 4. Discussion

Recent advances in biomarker research have revealed significant insights into the diagnostic potential of serum-based markers for neurodegenerative diseases. In a groundbreaking investigation, Steinacker et al. [28] demonstrated that serum NF-L and tau protein levels showed parallel elevation patterns to those in the CSF. NF-L levels are especially elevated in patients with CJD and those with motor neuron diseases. Thus, NF-L may be useful as a differential diagnostic tool.

Zerr et al. [29] conducted a comprehensive comparative analysis of plasma NF-L and t-tau levels in healthy controls and patients with a diverse spectrum of neurological disorders, including non-neurodegenerative neurological diseases (both with and without dementia syndrome), AD, CJD, Lewy body disease (LBD), frontotemporal lobular dementia, and vascular dementia. The results were particularly striking in the CJD cohort; patients with CJD had the highest plasma NF-L and t-tau levels. These biomarkers exhibited remarkable discriminatory power in distinguishing CJD from other forms of dementia, with an area under the curve (AUC) of 0.93. The finding that plasma t-tau levels, but not plasma NF-L levels, were significantly correlated with disease duration was of particular clinical significance. Furthermore, t-tau exhibited moderate predictive capability for patient survival outcomes. These findings enhance our understanding of the pathophysiological mechanisms underlying CJD and provide promising implications for early diagnosis and prognosis prediction. Our analysis revealed no significant correlations between CSF and serum NF-L levels and various clinical parameters in patients with CJD. Specifically, we examined the relationships between NF-L levels and disease duration (Figure 1), the temporal progression from disease onset to the development of akinetic mutism (Figure 2), and age at disease onset (Figure 3). We detected no significant associations among these variables. The differences between our findings and previous studies’ findings may be due to the differences in treatment approaches between Japan and the West.

Iwasaki et al. described that the survival times of Japanese patients with sCJD are not significantly different from those of Western patients with CJD regarding the time that elapsed before they become akinetic and mutable [3]. Additionally, differences in life extension due to tube feeding and subsequent symptomatic treatment play a role in survival time [3].

Our results differ from North American and European study results, possibly because Japanese data cover a long period of akinetic and variable periods, making it difficult to establish correlations. Therefore, to further clarify the potential prognostic impact, we performed a stratified analysis by dividing the patient cohort into two different groups based on disease progression, including a rapid progression group (disease duration ≤ 20 months) and a slow progression group (disease duration > 20 months) (Figure 3). Even in this detailed subgroup analysis, we did not detect associations between disease progression patterns and NF-L concentrations in either CSF or serum samples. Contrary to our initial hypothesis, these results suggest that NF-L levels are not a reliable prognostic indicator of disease progression in patients with CJD. We also investigated the period from onset to the akinetic and variable periods; however, we did not detect any correlations during this period (Figure 2).

Another possibility was a difference in the measurement methods. Reports by Steinacker et al. [28] and Zerr et al. [29] utilized ultrasensitive single molecule array (Simoa^®^) assays. At the same time, our study employed the Simple Plex™ (Quanterix, Billerica, MA, USA) NF-L Assay on Ella^®^, which could have contributed to the different outcomes. Nevertheless, the values were comparable, making it improbable that the variation in measurement methods caused the differences in the results. Lastly, we considered the difference between serum and plasma. Plasma NF-L levels were analyzed in Zerr et al.’s study [29], whereas Steinacker et al. [28] focused on serum NF-L levels, and their findings were not significantly different. Given these outcomes, it is unlikely that the observed differences stemmed from using serum NF-L versus plasma NF-L. NF-L levels measured using the Ella^®^ platform were significantly correlated with measurements obtained using the Simoa^®^ immunoassay platform and Uman Diagnostics ELISA. Subsequently, the original sample concentrations were derived through retrospective calculation, incorporating the respective dilution coefficients (Appendix A).

The ultrasensitive single-molecule array (Simoa^®^) is the gold standard for CSF and serum NF-L quantification, as demonstrated in previous investigations by Steinacker et al. [28]. In contrast, the Ella^®^ automated immunoassay system was used for NF-L quantification in this study.

No significant differences were detected between our results and those previously reported by Steinacker et al. [28]. Measurements were performed using a smaller number of plasma samples, and the measurements were comparable to those in the study by Zerr et al. [29]. This high inter-method correlation provides solid evidence for the analytical reliability and validity of Ella^®^ in NF-L quantification. The agreement between our results and the results obtained using the established Simoa^®^ method validates the analytical accuracy and precision of the Ella^®^ system. This validation demonstrates the ability of the Ella^®^ system to provide reproducible and reliable results and the usefulness of this assay as an alternative analytical tool in clinical and research settings (Appendix A).

Based on all these factors, we determine that serum NF-L levels do not correlate with prognosis in Japanese CJD patients and are not viable biomarkers for this group. Moreover, this study observed no relationship between CSF or serum NF-L and survival time; however, a slight trend was identified connecting CSF NF-L and serum NF-L, albeit without statistical significance (Figure 1C).

As previously reported, the presence of several neurodegenerative disorders, including ALS, is significantly correlated with plasma and serum NF-L levels [17,18,19,20,21,22,23,24,25,26]. ALS is characterized by selective neuronal damage, but patients with CJD exhibit pathological changes in multiple cell types, including neurons, astrocytes, and microglia. Therefore, NF-L concentration alone may not be sufficient as a comprehensive indicator of disease progression in patients with CJD. Moreover, CJD typically progresses more rapidly compared with other neurodegenerative disorders, and the sampling frequency may not adequately reflect real-time disease progression. The ease of sampling is a crucial consideration when addressing this limitation. Considering the invasiveness of sample collection and potential infection risks, serum or plasma sampling presents a more favorable approach compared with CSF collection.

The sample size in this study is relatively small, highlighting the need to increase the number of samples in future research. Currently, ultrasensitive single molecule array (Simoa^®^) assays dominate NF-L measurement. However, in the future, many samples are expected to be analyzed using the Simple Plex™ NF-L Assay on Ella^®^™, a next-generation system that offers easier measurement. This could lead to variations in results between Europe, the United States, and Japan, as noted in this context, and more evidence is needed.

This research represents a significant advancement in neurodegenerative disease diagnostics, potentially offering less invasive alternatives to traditional CSF analysis. Reliable blood-based biomarkers could revolutionize both the diagnostic process and monitoring of disease progression, ultimately leading to improved patient care and management strategies.

## 5. Conclusions

We employed a fully automated multiplex ELISA (Ella^®^) to measure the concentrations of 14-3-3 protein, t-tau protein, and NF-L in CSF and serum samples from patients with prion disease and analyzed their link to disease prognosis. Although we did not confirm a correlation between NF-L levels and survival time in North American and European cases, our findings suggest that NF-L concentrations hold promise as a potential early diagnostic marker for CJD with further refinement. The potential impact of our findings on the early diagnosis of CJD patients is significant. Future research should focus on expanding the sample size of sporadic CJD cases in Japan and leveraging next-generation measurement techniques to validate and extend these preliminary observations.

## Figures and Tables

**Figure 1 biomolecules-15-00008-f001:**
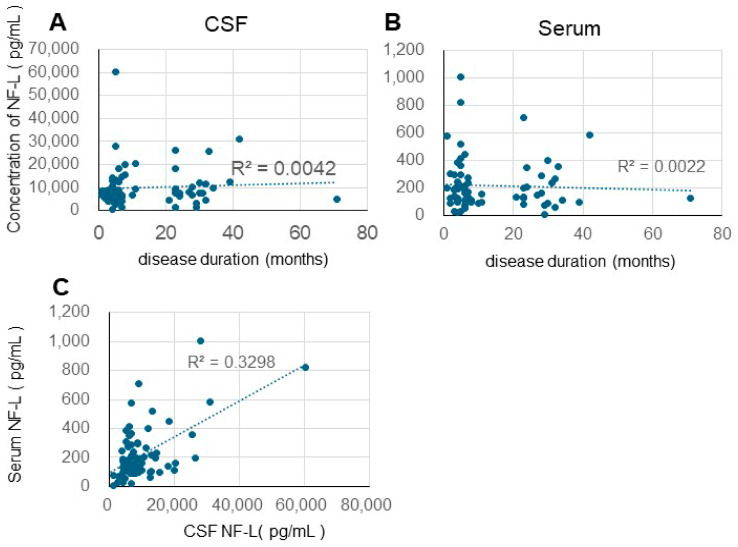
Correlation between disease duration and CSF/serum NF-L (**A**). Correlation between disease duration and CSF NF-L (R^2^ = 0.0042, Pearson’s CC = 0.0064) (**B**). Correlation between disease duration and serum NF-L (R^2^ = 0.0022, Pearson’s CC = 0.047) (**C**). Correlation between CSF and serum NF-L levels (R^2^ = 0.3298, Pearson’s CC = 0.576).

**Figure 2 biomolecules-15-00008-f002:**
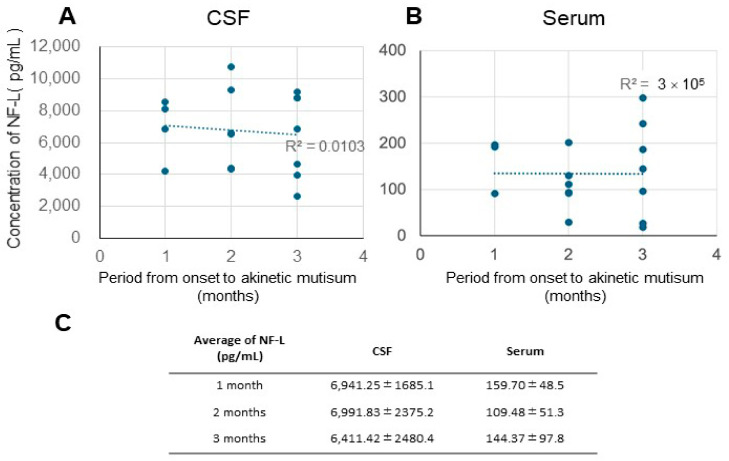
Correlation between duration of illness from onset to akinetic mutism and NF-L concentration (**A**), CSF c (**B**), serum (R^2^ = 3 × 10^5^, Pearson’s CC = −0.005) (**C**), and average NF-L in CSF and serum (median ± S.D.).

**Figure 3 biomolecules-15-00008-f003:**
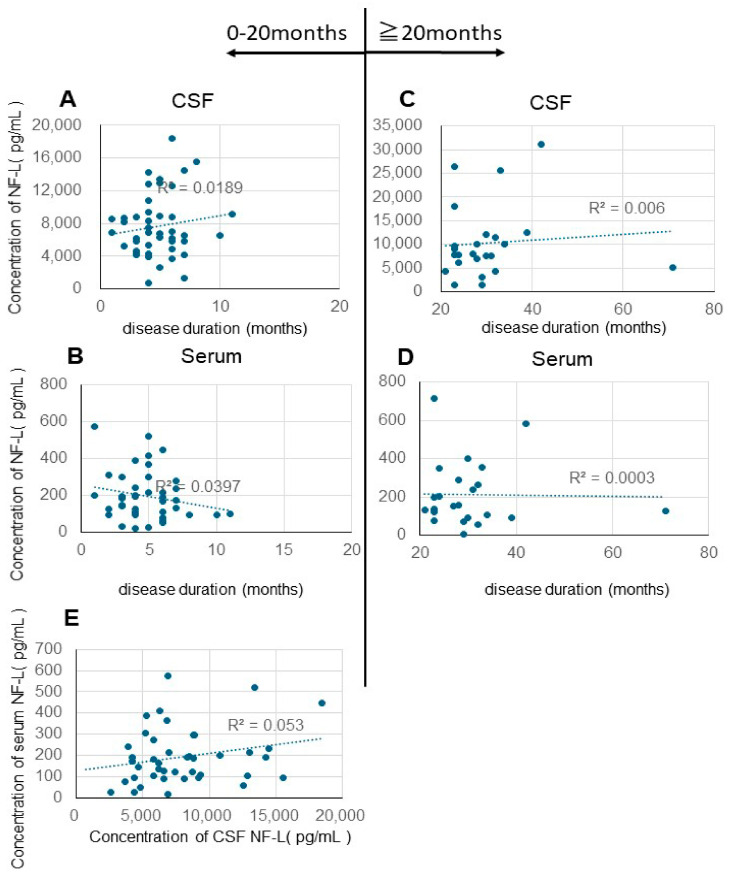
Correlation between cerebrospinal fluid NF-L/serum NF-L levels and disease duration (short-term vs. long-term). (**A**) and (**B**): 0-20 months (short-term). (**A**): R^2^ = 0.0189; Pearson’s CC = 0.137. (**B**): R^2^ = 0.0397, Pearson’s CC = −0.0199. (**C**) and (**D**) ≥20 months (long-term). (**C**): R^2^ = 0.006, Pearson’s CC = 0.077. (**D**): R^2^ = 0.0003; Pearson’s CC = −0.016. (**E**) Correlation between CSF NF-L levels and serum NF-L levels in the short term (0–20 months) (R^2^ = 0.053, Pearson’s CC = 0.2303).

**Figure 4 biomolecules-15-00008-f004:**
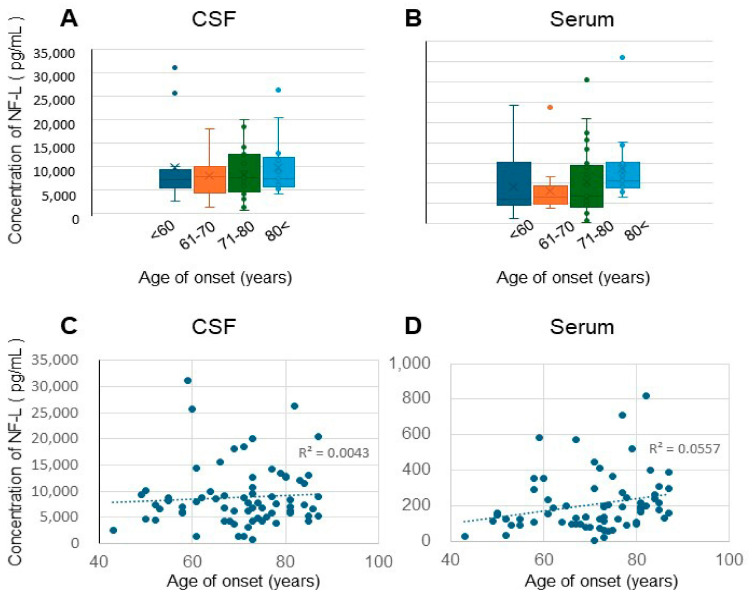
Relationship between age at onset and CSF NF-L. (**A**,**B**) Age at onset and NF-L concentration ((**A**): CSF; (**B**): Serum). (**C**,**D**) Correlation between age at onset and CSF/serum NF-L^©^, and correlation between age at onset and CSF NF-L (R^2^ = 0.0184, Pearson’s CC = 0.135). (**D**) Correlation between age at onset and serum NF-L (R^2^ = 0.0557, Pearson’s CC = 0.236).

**Table 1 biomolecules-15-00008-t001:** Summary of biomarker data in patients with Creutzfeldt–Jakob disease (CJD).

n	Age (Years) ^a^	Gender (f/m)	Serum NF-L (pg/mL) ^b^	CSF NF-L (pg/mL) ^b^	CFS Tau (pg/mL) ^c^ (≥1300 pg/mL)	14-3-3 Pos.	RT-QUIC Pos.
72	70.8 (43–87)	32/40	219.28 ± 184.5	9912.67 ± 8609.1	84.70%	88.9%	80.6%

f, female; m, male; pos., positive; ^a^ Age is given as average. ^b^ Markers are given as mean with standard deviation. ^c^ CSF total tau protein was considered positive (≥1300 pg/mL).

## Data Availability

The data were obtained from the patients’ attending physicians and the Japan Prion Disease Surveillance Committee at the time of the investigation and pathological autopsy.

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
