# Peer review of "Neurofilament Light Chain Levels in Serum and Cerebrospinal Fluid Do Not Correlate with Survival Times in Patients with Prion Disease"

_biomolecules, 2024, doi:10.3390/biom15010008_

Round 1
Reviewer 1 Report
Comments and Suggestions for Authors
Shimamura et al. have provided valuable insight into a lack of correlation between neurofilament light chain levels in serum and CSF and CJD progression in Japan, which is different from reports from other countries, and the potential reason for the difference in patient care. This study also suggests a limitation of using serum and CSF NF-L as a biomarker for CJD.
I have concerns about this manuscript that the authors should address before publishing.
1. NF-L level is known to increase in serum and CSF at a very early stage of neurodegenerative diseases. At the first sample collection (clinical onset), the disease has likely progressed to a stage at which NF-L secretion to the serum and CSF has already reached a plateau. Given that CJD is a rapidly progressing disease, the serum and CSF NF-L levels are likely to have reached a plateau phase. The authors have partially discussed this in the Discussion; however, a direct comparison to serum and CSF NF-L levels in a slow-progressing prion disease like Gerstmann-Straussler Scheinker disease (GSS) would help.
2. The authors claim that tube feeding and other aspects of patient care in Japan contribute to the lack of CSF/serum NF-L correlation with CJD progression in Japan. Were any of these patients put on therapeutic drugs known to reduce prions? This information likely affects the correlation reported here.
3. Is there any attempt to discuss why tube feeding would affect NF-L levels in patients from Japan? Are the patients well hydrated by tube feeding, and is this affecting the concentration of NF-L in the serum and CSF?
4. Using "Western countries" to represent countries outside Japan is misleading and limits the information this manuscript could share with the field. The authors should list all countries or regions to provide more general information that will help the field.
Author Response
Dear Reviewer 1:
We would like to thank the reviewer for their valuable suggestions and comments. We addressed all the issues raised by the reviewer. We hope that the revisions are satisfactory. The point-by-point responses are shown below.
- NF-L level is known to increase in serum and CSF at a very early stage of neurodegenerative diseases. At the first sample collection (clinical onset), the disease has likely progressed to a stage at which NF-L secretion to the serum and CSF has already reached a plateau. Given that CJD is a rapidly progressing disease, the serum and CSF NF-L levels are likely to have reached a plateau phase. The authors have partially discussed this in the Discussion; however, a direct comparison to serum and CSF NF-L levels in a slow-progressing prion disease like Gerstmann-Straussler Scheinker disease (GSS) would help.
We are grateful for the suggestion. We are fully aware that the present study focuses on sporadic Creutzfeldt–Jakob disease. Although we have CSF samples from patients with GSS, we do not have blood samples. Moreover, the number of cases is too small to be included in this paper. We believe that our study can be further developed by increasing the diversity of our sample and comprehensively investigating a wider range of neurodegenerative diseases, which would greatly enhance the scientific significance and academic value of this important study. We are grateful for the opportunity to develop our work in future studies.
- The authors claim that tube feeding and other aspects of patient care in Japan contribute to the lack of CSF/serum NF-L correlation with CJD progression in Japan. Were any of these patients put on therapeutic drugs known to reduce prions? This information likely affects the correlation reported here.
Thank you very much for your thoughtful comment. Therapeutic drugs were not used in any cases in this study.
- Is there any attempt to discuss why tube feeding would affect NF-L levels in patients from Japan? Are the patients well hydrated by tube feeding, and is this affecting the concentration of NF-L in the serum and CSF?
We would like to address several important considerations regarding our research findings. First, we agree that adequate patient hydration through tube feeding or enteral nutrition is not likely to significantly influence the neurofilament light chain (NF-L) levels observed in our Japanese patient cohort.
We primarily focused on two critical temporal measurements: the duration from disease onset to mortality and the duration from disease onset to patient immobility, alongside their corresponding NF-L values. In our comparative analysis, we observed notable distinctions between European and Japanese patient populations.
Although European patient data demonstrated a correlation between the duration of disease onset to mortality and plasma NF-L levels, our Japanese cohort presented a distinct profile. Interestingly, we found no statistically significant correlation between NF-L levels and the duration from disease onset to mortality in Japanese patients. We hypothesize that this divergence may be attributable to several factors, with tube feeding potentially representing a significant confounding variable.
- Using "Western countries" to represent countries outside Japan is misleading and limits the information this manuscript could share with the field. The authors should list all countries or regions to provide more general information that will help the field.
Thank you for this valuable feedback. We have replaced “Western countries” with “North American and European cases.”

Reviewer 2 Report
Comments and Suggestions for Authors
The authors have prepared a manuscript detailing an analysis of the neurofilament subunit L (NF-L) concentration in patients suffering from spontaneous Creutzfeldt-Jakob disease. This is compared with other diagnostic biomarkers (T-tau, 14-3-3, RT-QuIC) generated using the same set of patient samples from serum and CSF, as detailed in a prior publication. Overall, the authors found that NF-L levels do not correlate with overall patient survival time, which is in contrast to the earlier findings of groups based in the Western world. The authors suspect that this may be due to differences in end-of-life care in each of these societies.
This finding is indeed interesting and may have diagnostic implications for patient care. Overall, the manuscript is well constructed and follows a logical flow that makes reading and understanding quite easy. I have a few minor concerns, however. First, the introduction is somewhat under-cited in general, and specifically in regard to the differences between Western and Eastern practices in end-of-life care. For instance, the paragraph starting with "In other hand..." should have at least one citation supporting the claims therein.
Next, there are a few issues with the methods section. Are the PrP genotypes of the patients available? Also, though these are all listed as sCJD cases, is there any information on the CJD subvariant each patient presented with? Regarding the actual analytical methods, it would be helpful if the authors could include manufacturer's part numbers, particularly for the NF-L kit that was used. Lastly, information regarding the statistical methods employed are severely lacking. The authors state that 'simple regression' was used. Is this a least-squares method, or some other methodology? Further, was software used in this analysis, or was it conducted solely by hand? Inclusion of these details would greatly improve the manuscript.
Comments on the Quality of English LanguageOverall, the English in the manuscript is good, but there are a few errors throughout. It would serve the authors well to have the manuscript re-read by a native speaker or editing service to smooth out these issues. A few examples I noticed include the sentences "However, Western countries..." and Future research should focus..." in the abstract. In the introduction, the sentence "Elevated levels of these markers..." is an incomplete sentence. Other examples can be found in various places as well, but again, these are minor and should be fairly easy to address.
Author Response
Dear Reviewer 2:
We would like to thank the reviewer for their valuable suggestions and comments. We addressed all the issues raised by the reviewer. We hope that the revisions and explanations are satisfactory. The point-by-point responses are shown below.
First, the introduction is somewhat under-cited in general, and specifically in regard to the differences between Western and Eastern practices in end-of-life care. For instance, the paragraph starting with "In other hand..." should have at least one citation supporting the claims therein.
Thank you for the suggestion. We have increased the number of references.
Next, there are a few issues with the methods section. Are the PrP genotypes of the patients available? Also, though these are all listed as sCJD cases, is there any information on the CJD subvariant each patient presented with?
The genetic tests were already performed and no genetic mutations were detected. The codon 129 polymorphism for solitary prion disease was methionine-methionine. These results are listed in the Patient Profiles subsection of the Results.
Regarding the actual analytical methods, it would be helpful if the authors could include manufacturer's part numbers, particularly for the NF-L kit that was used. Lastly, information regarding the statistical methods employed are severely lacking. The authors state that 'simple regression' was used. Is this a least-squares method, or some other methodology? Further, was software used in this analysis, or was it conducted solely by hand? Inclusion of these details would greatly improve the manuscript.
Thank you for this valuable feedback. The manufacturer part numbers for NF-L have been added. Statistical processing was performed using JMP 17.2.
Comments on the Quality of English Language
Overall, the English in the manuscript is good, but there are a few errors throughout. It would serve the authors well to have the manuscript re-read by a native speaker or editing service to smooth out these issues. A few examples I noticed include the sentences "However, Western countries..." and Future research should focus..." in the abstract. In the introduction, the sentence "Elevated levels of these markers..." is an incomplete sentence. Other examples can be found in various places as well, but again, these are minor and should be fairly easy to address.
Thank you for pointing this out. We apologize for the poor English. The manuscript has now been carefully reviewed by an experienced editor whose first language is English and who specializes in editing manuscripts written by scientists whose native language is not English.
